# Local Strain Effects on Lattice Defect Dynamics and Interstitial Dislocation Loop Formation in Irradiated Tungsten–Molybdenum Alloys: A Molecular Dynamics Study

**DOI:** 10.3390/ijms251910777

**Published:** 2024-10-07

**Authors:** Marzoqa M. Alnairi, Mosab Jaser Banisalman

**Affiliations:** 1Department of Physics, Umm Al-Qura University, Makkah 24382, Saudi Arabia; 2EN2CORE Technology, Daejeon 34127, Republic of Korea

**Keywords:** primary radiation damage defects, strain effects, tungsten, tungsten–molybdenum, molecular dynamics, collision cascade, dislocation loops

## Abstract

In this study, molecular dynamics (MD) simulations were used to investigate how alloying tungsten (W) with molybdenum (Mo) and local strain affect the primary defect formation and interstitial dislocation loops (IDLs) in W–Mo alloys. While the number of Frenkel pairs (FPs) in the W–Mo alloy is similar to pure W, it is half that of pure Mo. The W–20% Mo alloy, chosen for further analysis, showed minimal FP variance after collision cascades induced by primary knock-on atoms (PKAs) at 10 to 80 keV. The research examined hydrostatic strains from −1.4% to 1.6%, finding that higher strains correlated with increased FP counts and cluster formation, including IDLs. The following two types of IDLs were identified: majority ½ <111> loops as well as <100> IDLs that formed within the initial picoseconds of the simulations under higher tensile strain (1.6%) and larger PKA energies (80 keV). The strain effects also correlated with changes in threshold displacement energy (TDE), with higher FP formation under tensile strain. This study highlights the impact of strain and alloying on radiation damage, particularly in low-temperature, high-energy environments.

## 1. Introduction

Tungsten (W) and its alloys, particularly when alloyed with molybdenum (Mo), are pivotal refractory metals that have carved out a significant niche in both nuclear and high-temperature applications [1,2]. Tungsten’s inherent strength and high melting point combined with the unique properties of molybdenum, render W–Mo alloys remarkably resistant to wear, corrosion, and heat [1,3,4]. The W–Mo alloys exhibit robust tensile strength at room temperature and maintain commendable ductility under extremely cold conditions, such as at −196 °C. This impressive material performance positions tungsten ahead of several other refractory metals in terms of durability and resilience.

Furthermore, tungsten’s ability to form a protective oxide layer ensures its resistance to corrosion from a plethora of acids and liquid metals, especially at temperatures under 150 °C. It is, however, crucial to be cautious of oxygen contamination during high-temperature processes like low vacuum annealing [1] or welding, due to its affinity for oxygen. While tungsten on its own offers significant advantages, the emergence of W–Mo alloys has garnered considerable attention in the realm of high-Z plasma-facing materials for future fusion reactors. Recent studies underscore tungsten’s superiority over other refractory metals, noting its higher resilience to He+ ion-induced surface nano-structuring compared to metals like tantalum. This crucial property underscores a reduced risk of reactor contamination from material degradation [5]. Furthermore, by combining tungsten and molybdenum, one might anticipate improved mechanical robustness, enhanced thermal stability, and even the potential mitigation of tungsten’s oxygen-related challenges.

Utilizing W and W-based alloys has proven advantageous, particularly in generating focused neutrons under spallation conditions. When exposed to proton energy of 1 GeV, a tungsten target can spall more than 18 neutrons, a yield on par with other heavy metals such as tantalum, mercury, and lead [6]. This efficiency positions tungsten and its alloys as prime candidates for both spallation neutron source targets and target cladding material [7]. However, a notable innovation in material selection is the use of TZM as the core absorbing material. This choice is predicated on the enhanced applicability found when molybdenum alloys are combined with tungsten [8], in contrast to materials like tantalum alloys, which align better with tungsten for cladding purposes.

W–Mo alloys are known for their proven performance under rapid deformation [9,10]. Studies have shown that W–Mo alloys with 10–15 wt.% Mo have superior mechanical properties, including increased strength and plasticity at temperatures up to 1700 °C, compared to pure W [11]. The addition of Mo forms a solid solution with W, resulting in a stable bcc lattice structure and enhanced ductility, as indicated by a higher B/G ratio and Poisson’s ratio [12]. Although the mechanical strength of W–Mo alloys is not as high as pure W, it significantly exceeds that of pure Mo. Additionally, alloying W with Mo greatly improves oxidation resistance [13]. Incorporating W or Mo into Nb–Ti–Ni alloys has also been found to reduce hydrogen solubility and increase resistance to hydrogen-induced embrittlement [14].

Building on the discussion of W–Mo alloys, these materials can also be engineered to enhance properties such as irradiation resistance and reduced brittleness. In fusion reactors, where materials endure intense heat flux and neutron exposure, helium accumulation can cause expansion and increased fragility in unalloyed metals. Therefore, developing advanced alloys, such as W–Mo, is crucial for improving the performance and safety of fusion and fission reactor systems. Several studies have highlighted the potential of W–Mo alloys for use in irradiated environments due to their promising characteristics [15,16]. When a material is exposed to an irradiated environment, it encounters various energetic particles. The collision of high-energy particles such as protons, neutrons, electrons, and ions with the atoms in the material generates primary knock-on atoms (PKAs) with significant energy. These PKAs trigger a cascade of collisions, leading to the formation of localized defects, either as individual point defects or as extended line defects like dislocations. Over time, these collisions can result in the accumulation of defect clusters, void formation, and the creation of stacking faults, all of which contribute to the degradation of the material’s structural integrity [17,18].

When assessing the integrity of a structural material, dislocations are a crucial type of defect to consider [19,20,21,22,23]. Interstitial dislocation loops (IDLs) form when the material deforms in response to external stressors such as exposure to radiation. These IDLs serve as barriers, hindering dislocation motion via the Orowan mechanism, thereby enhancing the hardness of the material. However, IDLs are typically unwanted defects because they contribute to various degradation processes, including irradiation-induced creep, material swelling, and other issues that can adversely affect a system’s long-term stability and safety.

IDLs tend to emerge along two specific Burgers vectors, ½ <111> and <100> [24,25,26]. Studies have indicated that IDLs form under conditions of relatively high PKA energy, for example, at energies around 150–180 keV in pure W [27] and in Mo, respectively [28]. Although the structural and mechanical responses of pure W have been studied extensively, there is a dearth of in-depth investigations into the influence of collision cascades and strain on pure tungsten and its alloys [29,30,31], particularly under irradiation conditions.

In the context of reactor operation, materials are subjected not only to collision cascade events but also to various strain sources, such as the one due to the expansion of irradiation-induced voids and the strain from solute segregation, commonly referred to as “local strains.” These local strains play a crucial role in shaping our comprehension of defect evolution and exert a substantial effect on the quantity, type, and configuration of the defects that form. Consequently, these factors are key determinants of the material’s properties [32].

Molecular dynamics (MD) simulations have been employed to examine cascades in tungsten and its alloys, aiming to comprehend the cascade dynamics and defect formation [33,34,35,36,37,38,39].

Although it has been proposed that energetic cascade events might generate highly mobile IDLs in certain refractory materials, evidence from experiments or MD simulations specifically supporting this in strain-free W–Mo alloys is lacking. As a result, more investigations are essential to elucidate the influence of radiation and local strains on such materials and to formulate measures to counteract their joint effects. A suggested research involved exploring W–Mo alloys, focusing on displacement cascades in both the pure W and 50:50 alloy compositions, with recoil energies reaching up to 20 keV and focusing on the physical parameter’s effects under the strain-free condition [40].

It is also important to highlight that research on W–Mo alloys has been relatively limited compared to studies on W alone. Therefore, in our research, we utilized MD simulations to explore how local strain influences defect evolution during high-energy collision cascades in W–Mo alloys. Our primary objective is to decipher the processes leading to the formation of IDLs and to understand the impact of strain. We subjected samples to six distinct levels of hydrostatic strain (ranging from −1.4% to 1.6%) and exposed them to various PKA deposition energies (10 keV, 30 keV, 60 keV, and 80 keV). Through examining the formation, durability of the Frenkel pairs, and the aggregation of self-interstitial atoms (SIAs) under varying strain scenarios, along with analyzing IDLs, we aim to deepen our comprehension of radiation-induced damage mechanisms in these materials.

## 2. Results and Discussion

### 2.1. The Effect of Mo Alloying Ratio on the Survived Frenkel Pairs in W

The deviation in the number of FPs created was incorporated using the standard error of the mean (SEM), which is calculated as the average of SEM = σ/√n, where σ is the standard deviation of the average number of FPs over the set of *n* = 16 separate simulations for three different W–Mo configurations. Initially, Figure 1a demonstrates the evolution of damage over time as simulated through MD for both pure and alloyed systems at a PKA energy of 30 keV. This reveals that each type of structure follows a similar pattern of a thermal spike across all PKA energy levels. The kinetic energy imparted by the PKA leads to a rapid rise in atomic displacements during the initial, ballistic phase, culminating in the formation of the maximum number of defects before a significant portion of these defects return to their original positions within the crystal structure, a phenomenon referred to as recombination or stabilization. A key feature of this phase is the peak time, defined as the duration from the initiation of the event to the point where the defect count is at its highest. Analyzing peak defects is key to understanding the initial radiation impact and defect formation, while studying the survived defects reveals material resilience, long-term stability, and self-healing dynamics post-radiation exposure.

The effect of Mo alloying resulted in greater peak heights and longer periods before stable recombination especially for 10, 20, and 30%, while the pure structures led to earlier peaks, lower values, and shorter periods, with the lowest peak for pure W. It is notable that the peak counts vary considerably between the alloyed and pure forms. This variation likely leads to different evolutions of FPs, particularly in terms of clustering patterns. Secondly, the number of surviving FPs varies with changes in the Mo content within the W–Mo alloys. However, when compared to the pure W structure, the number of surviving FPs in the alloyed W–Mo remains similar, with a variation of approximately ±5 FPs (Figure 1b). This number typically falls within the 30 s range. In stark contrast, pure Mo deviates significantly from this trend, exhibiting around 70 survived FPs. While these results align to some extent with those reported in studies utilizing the Gaussian approximation potential for W–Mo alloys [40], these observations underscore the complex interplays between the alloy composition and the defect dynamics, highlighting the nuanced behavior of these materials under similar conditions.

Additionally, variations in the PKA energy levels led to discrepancies in the volumes of the cascade cores and the quantities of defects generated [41]. With an increase in PKA energy (as shown in Figure 1c), the defects experienced more frequent collisions, causing the energy to spread across a broader area. This scenario reduced the chances of defect recombination, thereby escalating the defect count. This process induced a cycle where increased collisions resulted in the generation of additional defects, especially when the PKA energy surpassed the TDE [17]. Conversely, a reduction in the PKA energy meant that the atoms had less energy, travelled shorter distances, and moved more slowly away from the cascade core, resulting in a diminished production of defects. At an energy level of 80 keV, we observed multiple peaks before recombination occurred. These peaks are the result of various sub-cascade events that happened along the collision path of the PKA. When the secondary atoms collided, they gained energy from the initial impact. This led to independent sub-cascades that occur after the primary and main cascade.

Utilizing optimized atomic structures, we assessed the elastic properties of the W–Mo binary alloys, subsequently calculating their elastic modulus and other mechanical characteristics. The findings indicated that while the W–Mo alloys exhibited lesser mechanical robustness compared to pure W, their strength significantly exceeded that of pure Mo. Moreover, an analysis of the mechanical properties, specifically the B/G ratio and Poisson’s ratio, revealed that the Mo addition noticeably enhanced the ductility of pure tungsten. The incorporation of Mo contributed positively to the strength and ductility of W [42], encouraging grain size refinement and strengthening the alloy through the selective precipitation of W–Mo clusters [43]. Furthermore, Mo addition lowered the free-energy change in the W–Mo–Cu ternary system thus optimizing the alloying process [44]. The higher melting point of Mo also acted as a sink for interstitials and vacancies, reducing defect formation and enhancing the alloy’s resistance to irradiation hardening [45].

Overall, the addition of Mo to W enhanced its mechanical properties, thermal stability, and alloying behavior, making it suitable for various applications, including aerospace [42].

### 2.2. The Influence of Displacement Direction and Strain on the Formation of Survived Frenkel Pairs

The generation of initial radiation damage is a multifaceted process that is influenced by numerous factors such as the anisotropic structure of the crystal, the application of strain, and the disorderly nature of collision events, which include the formation of sub-cascades and channeling [46]. The random nature of the cascade mechanism presents challenges in predicting the redistribution of energy from the PKA, therefore requiring the adoption of a larger set of statistical data. To investigate the influence of PKA energy and direction on the formation of radiation damage, 16 separate simulations of displacement cascades were performed for each PKA recoil energy within each strained structure. The outcomes were assessed based on the number of survived FPs, which are the defects that remained after the relaxation period, as depicted in Figure 2.

The radiation damage process involves a complex interplay of factors, including the crystal’s anisotropic configuration, the application of strain, and the chaotic nature of collision cascades that lead to sub-cascade creation and channeling effects [46]. The unpredictability of cascade reactions complicates the task of forecasting how energy is redistributed by PKA, highlighting the need for extensive statistical data. To delve into how the PKA energy and its direction impact radiation damage creation, 16 unique simulations of displacement cascades for each PKA recoil energy were executed across differently strained structures. The evaluation focused on the count of FPs that survived the relaxation period, as shown in Figure 2.

Figure 2a,b display the representation of the influence of various PKA displacement directions, namely <111>, <110>, <100>, and <321>, on the survival of the FPs over various alloying ration and as per PKA changes. It is evident that the outcomes differ in different directions, with the most deviation in the number of FPs observed with pure Mo, 10% Mo, and pure W, while the lowest deviation was with the alloying of Mo (either 20 or 30% of Mo). The data show a hierarchy in the crystallographic directions, for example, for the case of pure Mo, with <111> being greater than <100>, which in turn is greater than <110> and <321>. This is corroborated by the evaluated TDE in these directions for both the pure Mo and pure W. However, this trend significantly changes when the structure is alloyed. The reasons for this alteration are multifaceted, as the TDE evaluation statistics undergo wide-ranging changes. These changes can be attributed to the collisions involving either the W or Mo atoms and the variations in the atomic ratios between these two elements. In contrast, examining the W–20 Mo alloy revealed that increasing the PKA energy did not significantly alter the trend of observing the highest defect counts in the <321> direction and the lowest in the <100> direction. However, the disparity in FP counts between these directions markedly increased with higher PKA energy. Specifically, the counts of FPs in the <321> direction sharply escalated at 80 keV compared to lower PKA energies, whereas the FP count in the <100> direction increased linearly and more gradually.

Once strain was applied, another key effect emerged in the W–20 Mo alloy. As we moved from compression to tension, the number of observed FPs increased (Figure 3a). This correlated with changes in the TDE, which also increased from compression to tension in the alloy (Figure 3b). Lower TDE values facilitated the easier formation of FPs, with less energy required to initiate them. Conversely, local compression raised the TDE, leading to fewer FPs being formed. Notably, this effect was most pronounced when the PKA energy was at 80 keV and the alloy was under 1.6% tensile strain, which resulted in a significantly steep increase in peak FPs.

The changes in the count of surviving FPs was more marked with increasing PKA energy levels, consistent with the observations made in the earlier research [41,47]. Additionally, the alloy demonstrated increased stability at a 1.6% strain level. Notably, there was a sharp increase in the number of surviving FPs, especially at PKA energies of 60 and 80 keV. This increase can be attributed to several factors, which will be analyzed in the subsequent sections. These factors primarily include the onset of clustering and the formation of interstitial dislocation loops under these conditions.

### 2.3. The Influence of Strain on Cluster and Dislocation Loop Formation

#### 2.3.1. Cluster Analysis

In the examination of defects resulting from irradiation, both the proportion of point defects clustering and the distribution of cluster sizes at the cascade’s conclusion within W–20 Mo alloys were investigated. Figure 4 illustrates the following: (a) the average number of interstitial clusters and (b) the number of vacancy clusters assessed. As the structural stress shifts from compression to tension, there is a corresponding change in the quantity of clusters formed within these materials. Predominantly, smaller clusters formed by interstitial atoms, referred to as SIA clusters and typically comprising two to four atoms, are observed. It was noted that clusters made up of a few vacancies or SIAs, typically fewer than three atoms, are common among both the SIA and vacancy cluster types.

Both the SIA and vacancy clusters have relatively comparable values over the tested strains, with the highest recorded value when the SIA clusters are under tensile strain and when the vacancy clusters are under compression strain. The clear distinction between the vacancy and SIA clusters seems to be more apparent for the SIA clusters as compared to the vacancy clusters when the system strain is 1.6%.

Furthermore, it was noted that the proportion of interstitial clusters relative to the overall cluster formation in the alloy systems exceeds that of the vacancy clusters by 0.4, aligning with the reported results in the previous research on the W–Re [48], Fe–Cr [49,50], and Ta–W alloys [51].

It is apparent that while vacancy clusters can form in larger groupings compared to interstitial clusters, their appearance in such extensive sizes occurs less frequently. The lower energy threshold for creating vacancies, as opposed to SIAs, supports the more frequent formation of vacancy clusters over SIA clusters. Observations indicate a maximum observed size of 40 for vacancy clusters and 17 for SIA clusters. Despite the more common occurrence of SIA clusters, they are found in much smaller numbers relative to other cluster sizes (e.g., 2, 3). Additionally, the growth rate of the interstitial cluster numbers is more gradual than that of the SIA clusters when under the influence of strain. In both the pure and alloyed systems, a higher total of clusters is seen under the conditions of tensile strain compared to scenarios of no strain or compressive strain, with the alloyed configurations showing a reduced cluster count under the latter conditions.

#### 2.3.2. Formation of Interstitial Dislocation Loops

Investigations into defect formation and clustering highlighted notable increases in the defect quantities in specific areas when the material experienced tensile strain. Further analysis using the DXA [52,53] allowed for a deeper exploration of how these defects come about as well as an assessment of the potential for IDLs to form.

In the study of W–20 Mo alloys, it was observed that the following two types of IDLs formed at a PKA energy of 80 keV: the ½ <111> IDLs and the <100> type. At lower PKA energies of 30 and 60 keV, only a small number of ½ <111> IDLs were detected, while even lower energies, such as 10 and 20 keV, did not result in the formation of any IDLs in the alloy, regardless of the strain levels applied (Table 1). Additionally, the IDLs identified within the alloyed framework were notably shorter and only manifested under tensile strains of 1.0% and 1.6%. However, the larger tensile strain led to lengthier IDLs. As per the displacement direction the <110> led to no chance of the IDL formation as compared to the other direction where the <100> direction was found to lead the formation of the longest IDLs and a higher number of segments, which was seven segments for ½ <111> compared to four for the <111> direction. The observation that no IDLs were formed in the <110> direction contradicts the expectation set by its lower TDE compared to the <321> direction, which did show relatively more IDL formation. This suggests that the formation of IDLs is not solely determined by the TDE value but also involves other complex factors that are not straightforward to explain in this context. However, when considering the <100> IDL formation type, this type is generally less common to occur; however, it formed through the <111> and the <100> displacement directions, indicating a nuanced interplay of the directional and energy factors in IDL formation in the W–20 Mo alloys. To gain insights into how the IDLs develop in both pure and alloyed structures, Figure 5 details our findings on the IDLs under a 1.6% tensile strain condition. We noted that the IDLs tended to form at the peak stage of tensile strain within the cascade and remain as such through to the stage of recombination.

This observation aligns with earlier studies that have documented similar patterns of sub-cascade activities in W alloys [54] and other metals like Fe [55]. These findings suggest that local strain conditions might lower the energy threshold needed for the formation of IDLs. For example, an IDL composed of 12 SIAs has an energy that is 9 eV less than the combined energy of 12 individual mono defects in pure W, as shown in the lower right section of Figure 5. This observation supports the study’s conclusion that different types of IDLs can form not only at very high PKA energies or with larger PKA masses but also at relatively lower PKA energies under the conditions of high tensile strain. Building on the initial observations, it was noted that, in the materials examined, W interstitials tended to form in marginally larger clusters instead of isolated dumbbell configurations in the case of the alloy structures. This aligns with earlier research indicating that the binding energy between an Mo atom substituting in the lattice and a W–W <111> dumbbell is negative. Such a negative value suggests a repelling force that discourages the creation of mixed W–Mo dumbbells, a conclusion supported by various prior studies [56,57].

The influence of the PKA collision direction on the formation of interstitial dislocation IDL has been identified as notably significant. As indicated in Table 1 and Figure 5, IDLs originating from the <111> PKA direction tend to form longer loops, a result that may stem from what is known as the focused collision sequence effect.

In such sequences, atoms collide predominantly along a single direction, and the resultant pattern is relatively unaffected by changes in the atomic number density. Furthermore, IDLs oriented in the more complex <321> direction were observed to be lengthier than those in other directions, exhibiting a stronger reaction to tensile rather than compressive strain. This implies that they extend further and form more segments under tension, likely due to an increase in FP clustering into IDLs during the ballistic phase. These observations highlight the critical role of collision displacement direction in the study of W SIAs within IDLs. It is crucial to recognize that this investigation zeroes in on the localized configuration of a specific alloy and does not capture the alloy’s average behavior. While the distribution of FPs, clusters, and IDLs might vary with different configurations, the influence of strain on these distributions is anticipated to remain consistent across various setups.

#### 2.3.3. Extra Exploration on Point Defects and Their Formation Efficiency

Our analysis revealed that defect production efficiency generally declines with an increase in PKA energy, except at a strain condition of 1.6% (leading to a volume change of 4.8%), as depicted in Figure 6a. Nonetheless, at elevated PKA energies, the efficiency of defect production either stabilized or exhibited a slight uptick [58,59], potentially linked to the emergence of IDLs. Notably, at high PKA energies and under tensile strains, particularly at 30 keV and 1.6%, IDL formation contributed to a rise in defect numbers, a phenomenon detailed in Section 2.3.1. This augmentation in defect numbers, reflecting an enhanced efficiency in defect production for the alloyed composition, is highlighted in Figure 6a. However, data concerning defect production efficiency in the pure composition at the 1.6% strain level were omitted from the illustration due to structural deformation and efficiency values surpassing the graphical limits.

We evaluated the initial radiation damage in materials through a mixed methodology incorporating both theoretical frameworks and MD simulations. To gauge the efficiency of defect production, we compared the number of Frenkel pairs (FPs) projected by the NRT theoretical model [61] to the actual count of FPs identified in the MD simulations, calculating the ratio (N_FP_/N_NRT_). This comparison is illustrated in Figure 6b.

The outcomes of this research indicate that structures under irradiation should ideally not be exposed to tensile strain conditions, particularly at elevated PKA energies. Additionally, structures that possess local defects and impurities require careful management due to their heightened vulnerability to strain. The information gathered from this study, notably on the formation of IDLs, provides a valuable resource for conducting object kinetic Monte Carlo (OKMC) simulations on primary cascade damage in both pure tungsten and tungsten–molybdenum alloys.

## 3. Simulation Methods

All MD simulations were conducted using the Large-scale Atomic/Molecular Massively Parallel Simulator (LAMMPS) package [62].

The interatomic interactions were modeled using the embedded atom method (EAM) potentials, as first described by Chen et al. [43,63]. The parameters for tungsten were adjusted based on a combination of experimental data and first-principles calculations, whereas the parameters for the W–Mo interactions were specifically tailored using first-principles data.

The evaluation of the W–Mo parameters was carried out through density functional theory (DFT) calculations, and their comparison to other MD parameters was performed when necessary.

It has been demonstrated that this EAM potential has acceptable predictions for the threshold displacement energy (TDE) and defect energies [64,65,66].

MD simulations initiated by setting the simulation cell of a freely deforming structure to equilibrium under initial conditions of 30 K temperature and 0 Pa pressure. The baseline temperature for the recoil simulations was set at 30 K, although, for certain analyses, it was lowered to 0 K to eliminate the effects of atomic thermal vibrations. Simulations conducted at 30 K aimed to replicate the conditions similar to those in the initial experimental setups, denoted as 30 K recoil simulations [67]. In contrast, simulations at 0 K were designed to provide data that were unaffected by thermal vibrations. To account for the impact of thermal vibrations at 30 K on the TDE estimates [65], our approach involved conducting runs for each displacement direction at various times—0, 50, 100, and 150 fs. Averaging the outcomes of these simulations was strategic in reducing the effects of thermal vibrations and temporal correlations on the cascade outcomes.

The simulation framework for pure W involved creating 60 × 60 × 60 BCC supercells, encompassing a total of 432,000 W atoms. To simulate a W–Mo alloy composition, a specific number of W atoms within the supercell was substituted randomly with Mo atoms, thereby achieving a mixture with a defined Mo concentration. This process involved creating several random arrangements by varying which W atoms were replaced. For the purpose of increasing the reliability of the data pertaining to unstrained structures, three distinct random configurations were generated for each alloy system, with Mo concentrations of 5%, 10%, 20%, and 30%. In other words, for each concentration, three different structures were prepared to represent the same Mo ratio.

Consequently, the resulting average value can be considered as being based on quasi-random structures [68]. The PKA was selected as a W atom positioned at the center of the simulation box. Thus, for each Mo ratio, the results were derived from 48 simulations (4 different displacement directions ×4 timings ×3 random structures) for the alloy configuration and 16 simulations (4 displacement directions × 4 timings) for the pure W or Mo setup. The four displacement directions used in the simulations were <111>, <110>, <100>, and <321>.

The dimension of the simulation cell was carefully selected to ensure that the system’s average temperature did not surpass 200 K following the collision, even at the highest PKA energy levels. This approach aimed to avoid the expansion of displacement cascades beyond the simulation cell’s limits. Each system underwent testing under six different hydrostatic strain levels, spanning from −1.6% to 1.4%, with incremental adjustments of 0.6%. Hydrostatic strain was applied uniformly across all boundaries, where the negative values represented compressive strains and the positive values indicated tensile strains. The simulations adopted an adaptive timestep method, limiting the maximum displacement per step (x_max_) to 0.01 Å and the maximum timestep (t_max_) to 0.002 ps. This setup proved effective for observing defect formations across the various x_max_ and t_max_ parameters. The initiation of each recoil MD simulation involved assigning recoil energy to a PKA, calculated based on its velocity components. Defect formation within the lattice was assessed using Voronoi analysis [69]; a site was deemed to host an SIA if it contained more than one atom, identified as a vacancy if it was empty, and considered intact with no defect otherwise. For all the PKA energy levels, the induced displacement cascades were allowed to evolve over a 50 ps duration within a standard NVT ensemble (maintaining constant numbers of atoms, volume, and temperature), a period deemed sufficient to observe the cascade’s three ballistic phases (Table 2).

All simulations of cascade events were carried out in a micro-canonical (NVE) ensemble, maintaining the system’s atom count, volume, and energy constant throughout. Although the Norgett–Robinson–Torrens (NRT) model provides an estimate for atomic displacement, the main aim of conducting collision cascade simulations lies in quantifying the number of FPs produced through MD approaches [61].
(1)$$\mathrm{NRT}\;\mathrm{displacement}=\frac{0.8E_{\mathrm{de}}}{2E_{d}}$$

In this context, E_de_ signifies the nuclear energy deposition, closely presenting the energy imparted by the PKA for initiating the cascade effect. E_d_, on the other hand, represents the TDE for the material, identified as 90 eV for pure W and 62 eV for pure Mo [70], based on the current EAM and computational framework outlined in our study. Further details on the modeling approach and the calculated NRT displacements are presented in Table 1. The final atomic structures were examined using the OVITO software version 3.10.0, tailored for the post-processing and visualization tasks. Cascade dimensions and configurations were determined using various OVITO modifiers [71], while the identification of interstitial dislocation loops (IDLs) in the crystal matrix was facilitated by the Dislocation Extraction Algorithm (DXA) [52,53], which measures the Burgers vectors of IDLs and creates a visual line representation for analysis.

The findings derived from these simulations play a significant role in the identification of potential trends, such as the increase or decrease in the formation of Frenkel pairs (FP) and clustering, within both the free and strained conditions of the alloyed structures. The statistical reliability of these discoveries is substantiated by the methodology cited in [70].

The TDE was evaluated using a 12 × 12 × 12 simulation system, with each simulation consisting of 3456 atoms. This enabled the calculation of the TDE of the alloyed structures under different strain states as well. The TDE for the alloyed structures was subsequently determined by averaging the results of these simulations. This rigorous approach ensures a comprehensive and statistically robust assessment of the TDE, which is indispensable for comprehending the material’s response to displacement under diverse conditions. The evaluation of TDE of the alloy compositions is carried out following the procedure described in Equation (2) as follows:
(2)$$E_{d,avg}=\frac{1}{N_{ICD}}\sum_{i}^{N_{ICD}}E_{d,i}=\frac{1}{N_{ICD}}\sum_{i}^{N_{ICD}}\{\frac{1}{N_{timing}}\sum_{i}^{N_{timing}}E_{d,i,j}\}.$$
where *E_d,i,j_* signifies the threshold energy identified for recoil along the i-th direction during the j-th timing simulation, while *E_d,i_* represents the mean TDE calculated for recoil in the i-th direction. The term Ns is used to denote the count of distinct structures created randomly for varying tungsten content within the W–Mo alloy.

## 4. Conclusions

Through the application of molecular dynamics simulation, our study delved into the genesis of interstitial dislocation loops (IDLs) resulting from strain in both the tungsten–molybdenum (W–20 Mo) systems. Through the analysis of primary knock-on atom (PKA) energies of 10, 20, 30, 60, and 80 keV, we tracked the development of collision cascades at 30 K in structures subjected to volumetric (hydrostatic) strain across six different levels of strain. Our findings reveal an increase in the number of surviving Frenkel pairs (FPs) and self-interstitial atom (SIA) clusters as a result of structural deformations under tensile strain, whereas these numbers decrease under compressive strain. Notably, the W–20 Mo alloy system exhibited the lowest counts of FPs and SIA clusters, highlighting the impact of alloy composition and strain on defect formation and stability. Employing DXA analysis, we found that IDLs begin to aggregate at higher strain levels of 1.6%, initiated by a PKA energy of 80 keV. Our findings further reveal that strain induces a reduction in the barrier to IDL formation, thereby impacting material integrity. The detailed study of defect formation induced by strain in both the pure and alloyed W materials is critically important for guiding decisions on the use and dependability of these materials in vital applications, such as nuclear reactors and spacecraft. This research not only informs on material selection and design strategies for critical infrastructure but also plays a key role in driving the development of materials and technologies with enhanced resistance to radiation damage.

## Figures and Tables

**Figure 1 ijms-25-10777-f001:**
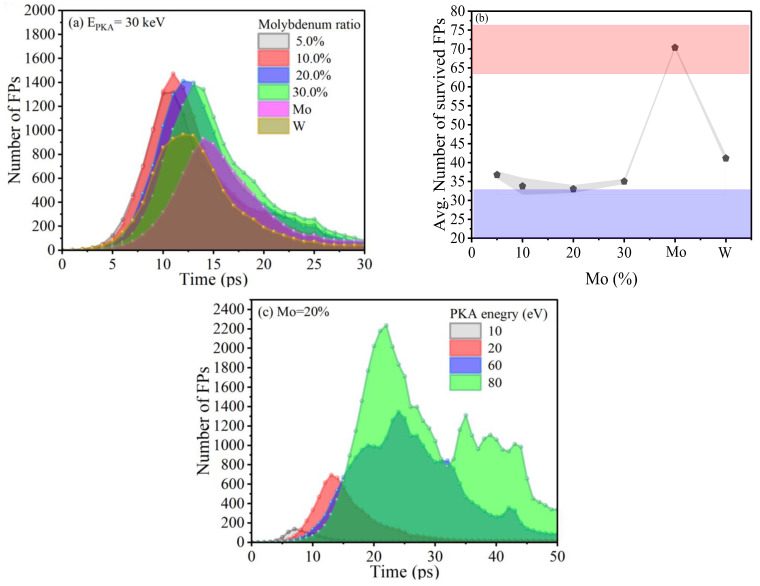
The diagram showcases the production of Frenkel pairs (FPs) throughout the collision cascades at 30 keV in (**a**) pure and alloyed structures at different Mo ratios (**b**) the survived number of FPs for different structures of alloyed and pure W (**c**) W–Mo alloyed material, with 20%Mo at PKA ranges from 10 to 80 KeV. Each data point aggregates the mean outcomes from four different directions and four separate timings, totaling sixteen unique samples.

**Figure 2 ijms-25-10777-f002:**
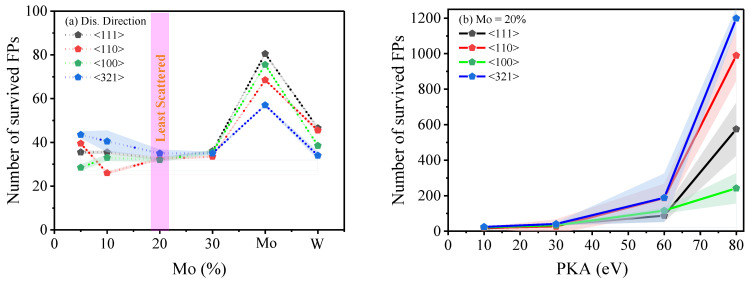
These graphs illustrate the surviving Frenkel pairs (FPs) when the 30 keV PKA projected over various recoil directions under different strain levels, focusing on (**a**) alloying effects of different Mo ratios and (**b**) a Mo-20% alloyed structure at different PKA energies. The shaded areas surrounding the plotted lines represent the standard error of the mean (SEM).

**Figure 3 ijms-25-10777-f003:**
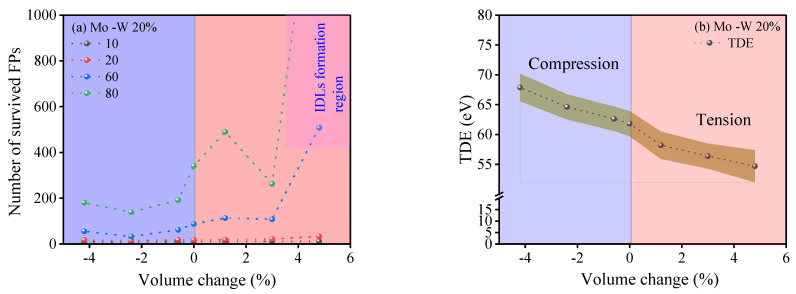
The graphs showcase (**a**) the projected maximum number of FPs across different PKA energy levels in relation to the level of hydrostatic strain applied and (**b**) the threshold displacement energy (TDE) for a W–20 Mo alloy in terms of defects remaining after 50 ps, with the pink area indicating where interstitial dislocation loops (IDLs) are likely to form. The shaded area around the strained graph represents the standard error of the mean (SEM).

**Figure 4 ijms-25-10777-f004:**
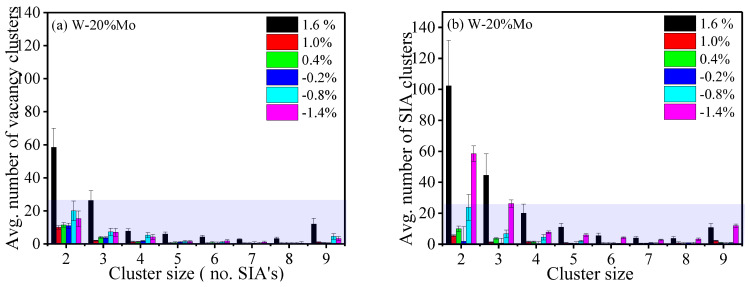
The graphs present the mean count of clusters formed, ranging from 2 to 9 self-interstitial atoms (SIAs) in size, across different levels of strain and at PKA energies of 80 keV. The average number of clusters is calculated over four varying directions of displacement, with the standard deviation illustrated. Part (**a**) focuses on SIA clusters, while part (**b**) deals with vacancy clusters.

**Figure 5 ijms-25-10777-f005:**
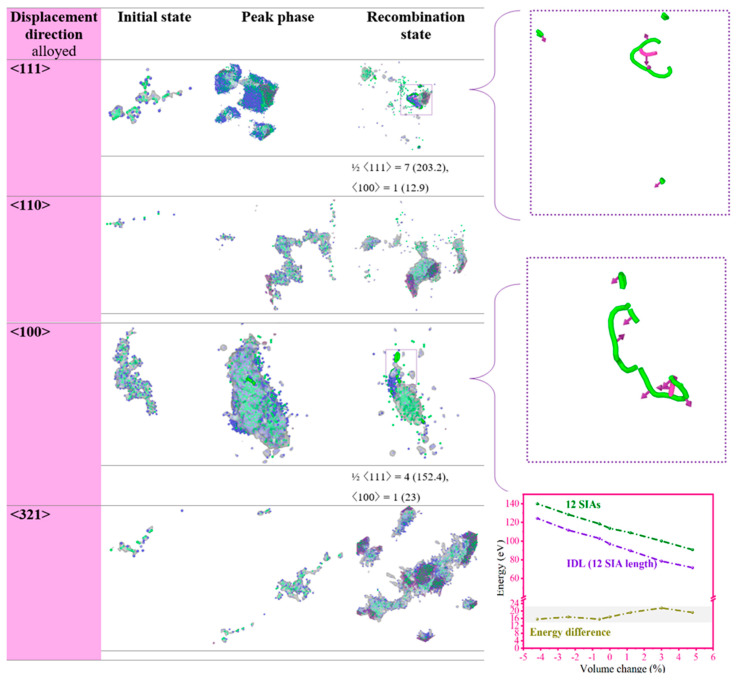
Displays the progression of FPs and the formation of IDLs across different directions of displacement. These simulations were conducted with a PKA energy of 80 keV under a strain of 1.6%, over a period of 30 ps. Blue atoms are shown as SIAs, green atoms indicate vacancies, light green loops depict ½ <111> IDLs, and pink loops illustrate the <100> loops. The lower left figure illustrates the correlation between the formation energy of single SIAs and IDLs as a function of strain variations.

**Figure 6 ijms-25-10777-f006:**
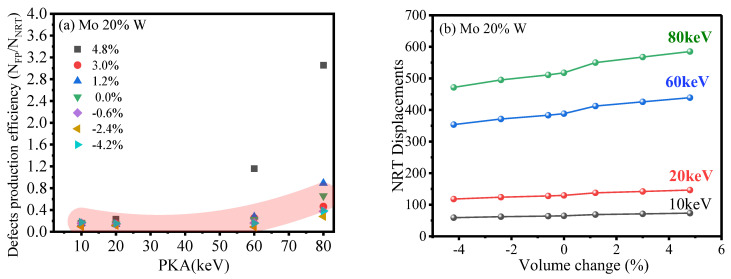
This figure displays the following: (**a**) the count of Frenkel pairs that persist across a range of PKA energies as influenced by applied hydrostatic strain, derived from the application of the NRT model with the formula V_NRT_ = 0.8 × E_pka_/2 E_d,j_(∆V) [60]; (**b**) the efficiency of defect production, quantified as the ratio between the outcomes from MD simulations and those predicted by the NRT model for defect calculations.

**Table 1 ijms-25-10777-t001:** The table displays a comparative analysis of Interstitial Dislocation Loop (IDL) formation in W–20%Mo subjected to 80 keV PKA cascades at various strain levels. This analysis extended across various strain levels and directions of collision cascades. Two types of IDLs are identified: Type A, which aligns with the ½ <111> type, and Type B, corresponding to the <100> type. The table includes the lengths of the IDLs, presented in parentheses. Additionally, a zero value in the table indicates conditions where IDL formation was not observed.

Displacement Direction at 80 KeV	−1.4%	−0.8%	−0.2%	0.4%	1.0%	1.6%
**<100>**	0	0	0	1A (101.5)	0	7A (203.2) 1B (12.9)
**<110>**	0	0	0	0	0	0
**<111>**	0	0	0	2A (44.5) 2B (42.5)	6 A (163.1)	4A (152.4) 1B (23)
**<321>**	00	1A (19.27)	0	0	2A (101.6)	0

**Table 2 ijms-25-10777-t002:** Simulation settings for collision cascades.

PKA Energy (keV)	System Count (Atoms)	System Size (Å)	Estimated NRT Displacements	Simulation Duration (ps)
5	432,000	16	33 (26)	50
10	432,000	16	66 (52)	50
15	432,000	16	99 (78)	50
30	432,000	16	198 (157)	50

## Data Availability

The simulations conducted in this study were performed using the LAMMPS package. Additionally, the structural design tools provided by Materials Square were employed. For those interested, Materials Square can be accessed at https://www.materialssquare.com, accessed on 15 February 2024.

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
