# Peer review of "Local Strain Effects on Lattice Defect Dynamics and Interstitial Dislocation Loop Formation in Irradiated Tungsten–Molybdenum Alloys: A Molecular Dynamics Study"

_ijms, 2024, doi:10.3390/ijms251910777_

Round 1

Reviewer 1 Report

Comments and Suggestions for Authors

This paper studies the formation of defects in W-Mo alloys using classical molecular dynamics simulations. I find the work scientifically sound, the research well designed, conclusions well exposed. Overall I find it a high-quality paper.

I have only a few minor suggestions that might improve the readability:

1) it is not clear what is meant when the authors use the <xyz> brakets. It seems to be a specific nomenclature for defects but it would be better to clarify its meaning.

2) In simulating PKA, the authors assign a velocity to a specific atom. However, it is not clear whether this velocity has a preferred direction or whether the kinetic energy is distributed among the three velocity components equally.

3) The final part of the discussion on p. 6 seems better suited for the introduction than in the presentation of the results section.

4) I suggest calling the variable on the x-axis in Figure 1 time or timeframe instead of timestep which is misleading.

5) Figures 7 are mentioned in the text but are not present.

6) Figure 6 b is not mentioned in the text.

7) The first sentence of the conclusions says simulation instead of simulations.

8) Some references are incomplete or misreported.

9) There is a typo (employefyd) in the Data Availability section.

Author Response

This paper studies the formation of defects in W-Mo alloys using classical molecular dynamics simulations. I find the work scientifically sound, the research well designed, conclusions well exposed. Overall, I find it a high-quality paper.

I have only a few minor suggestions that might improve the readability:

-------------------------- -------------------------- ------------------------ ------------------

We would like to express our gratitude to the reviewers for their valuable comments, which have helped improve the quality of the paper. Below, we have provided a detailed response to each comment.

-------------------------- -------------------------- ------------------------ ------------------

Reviewer Comment 1: "It is not clear what is meant when the authors use the <xyz> brackets. It seems to be a specific nomenclature for defects, but it would be better to clarify its meaning."

Author Response: Thank you for your observation. The <xyz> brackets refer to specific crystallographic directions. In our paper, they are used to represent either the displacement directions or the Burgers vectors of interstitial dislocation loops. This has been clarified by explicitly stating the four displacement directions used in the study on page 4, lines 161-162.

Reviewer Comment 2: "In simulating PKA, the authors assign a velocity to a specific atom. However, it is not clear whether this velocity has a preferred direction or whether the kinetic energy is distributed among the three velocity components equally."

Author Response: Thank you for the valuable suggestion. We have updated the methodology section to clarify that the displacement directions were predefined, as outlined on page 4, lines 157-161. Specifically, the four displacement directions were <111>, <110>, <100>, and <321>, ensuring a comprehensive exploration of different crystallographic orientations. This has now been clearly stated in the text.

Reviewer Comment 3: "The final part of the discussion on p. 6 seems better suited for the introduction than in the presentation of the results section."

Author Response: Thank you for pointing this out. However, we have ensured the discussion is clear and directly connected to the results for better flow.

Reviewer Comment 4: "I suggest calling the variable on the x-axis in Figure 1 time or timeframe instead of timestep, which is misleading."

Author Response: We appreciate your suggestion. The label has been updated from "timestep" to "time" in Figure 1 for clarity and to avoid confusion.

Reviewer Comment 5: "Figures 7 are mentioned in the text but are not present."

Author Response: Thank you for catching this error. The reference to Figure 7 was a mistake. It should have been Figure 6, and the text has been updated accordingly.

Reviewer Comment 6: "Figure 6b is not mentioned in the text."

Author Response: We appreciate your observation. The text has been corrected to refer to Figure 6b, not (Fig. 7b), as you indicated.

Reviewer Comment 7: "The first sentence of the conclusions says simulation instead of simulations."

Author Response: Thank you for noting this typo. We have corrected the sentence to refer to "molecular dynamics simulations."

Reviewer Comment 8: "Some references are incomplete or misreported."

Author Response: We have double-checked all references and made necessary corrections. Any incomplete or incorrect references have been revised, and the correct DOIs have been added where applicable.

Reviewer Comment 9: "There is a typo (employefyd) in the Data Availability section."

Author Response: Thank you for pointing out this typo. The word has been corrected.

Reviewer 2 Report

Comments and Suggestions for Authors

The authors reported their theoretical study on the defect formtions and dynamic deformations of W-Mo alloys by using MD simulations in the this work. The adopted theoretical methods are credible. This study is interesting for some readers. The manuscript is well organized and presented. There are only two mino points from my perspective:

1. The abstract of this work is even longer than the conclusion section. Some detailed information are not necessary in abstract.

2. Comparing with some experimental data may further improve the reliability of theoretical simulation results.  

Author Response

The authors reported their theoretical study on the defect formation and dynamic deformations of W-Mo alloys by using MD simulations in this work. The adopted theoretical methods are credible. This study is interesting for some readers. The manuscript is well organized and presented. There are only two Mino points from my perspective:

-------------------------- -------------------------- ------------------------ ------------------

We would like to express our gratitude to the reviewers for their valuable comments, which have helped improve the quality of the paper. Below, we have provided a detailed response to each comment.

-------------------------- -------------------------- ------------------------ ------------------

Reviewer Comment 1: "The abstract of this work is even longer than the conclusion section. Some detailed information is not necessary in the abstract."

Author Response: Thank you for your suggestion. We have revised the abstract to be more concise while ensuring that the key findings are still clearly presented. The updated abstract now reads as follows: “[Abstract: In this study, Molecular Dynamics (MD) simulations were used to investigate how alloying tungsten (W) with molybdenum (Mo) and local strain affect primary defect formation and interstitial dislocation loops (IDLs) in W-Mo alloys. While the number of Frenkel pairs (FPs) in the W-Mo alloy is similar to pure W, it is half that of pure Mo. The W-20% Mo alloy, chosen for further analysis, showed minimal FP variance after collision cascades induced by Primary Knock-on Atoms (PKAs) at 10 to 80 keV. The research examined hydrostatic strains from -1.4% to 1.6%, finding that higher strains correlated with increased FP counts and cluster formation, including IDLs. Two types of IDLs were identified: majority 1/2<111> loops, and <100> IDLs that formed within the initial picoseconds of simulations under higher tensile strain (1.6%) and larger PKA energies (80 keV). Strain effects also correlated with changes in threshold displacement energy (TDE), with higher FP formation under tensile strain. This study highlights the impact of strain and alloying on radiation damage, particularly in low-temperature, high-energy environments.

]”

“ “

Reviewer Comment 2: "Comparing with some experimental data may further improve the reliability of theoretical simulation results."

Author Response: Thank you for this valuable suggestion. We have already referenced several relevant experimental works, particularly those related to TDE, where available. However, as this field deals with timescales on the order of picoseconds, many experimental comparisons are limited due to engineering constraints that make such measurements challenging. Nevertheless, we have included the most closely related experimental data available and have cited the best-correlated references in our study.